# Disentangling Influence: Using disentangled representations to audit model predictions[*]

**Charles T. Marx**
Haverford College
cmarx@haverford.edu

**Richard Lanas Phillips**
Cornell University
rlp246@cornell.edu

**Sorelle A. Friedler**
Haverford College
sorelle@cs.haverford.edu

**Carlos Scheidegger**
University of Arizona
cscheid@cs.arizona.edu

**Suresh Venkatasubramanian**
University of Utah
suresh@cs.utah.edu

## Abstract

Motivated by the need to audit complex and black box models, there has been extensive research on quantifying how data features influence model predictions. Feature influence can be direct (a direct influence on model outcomes) and indirect (model outcomes are influenced via proxy features). Feature influence can also be expressed in aggregate over the training or test data or locally with respect to a single point. Current research has typically focused on one of each of these dimensions. In this paper, we develop *disentangled influence audits*, a procedure to audit the indirect influence of features. Specifically, we show that disentangled representations provide a mechanism to identify proxy features in the dataset, while allowing an explicit computation of feature influence on either individual outcomes or aggregate-level outcomes. We show through both theory and experiments that disentangled influence audits can both detect proxy features and show, for each individual or in aggregate, which of these proxy features affects the classifier being audited the most. In this respect, our method is more powerful than existing methods for ascertaining feature influence.

## 1 Introduction

As machine learning models have become increasingly complex, there has been a growing subfield of work on interpreting and explaining the predictions of these models [24, 11]. In order to assess the importance of particular features to aggregated or individual model predictions, a variety of *feature influence* techniques have been developed.

The goal of explaining model predictions is of particular importance in the context of fairness in machine learning. In human-centered prediction tasks such as recidivism prediction and consumer finance, understanding how protected attributes such as gender or race affect a prediction can improve transparency and aligns with the principles of contestable design [14]. While direct influence methods [7, 12, 20, 25] focus on determining how a feature is used directly by the model to determine an outcome, it is also possible for the model to access protected information through *proxy variables* – variables which are closely related the protected attribute. *Indirect feature influence* techniques [1, 2, 15] report that a feature is important if that feature *or a proxy* has an influence on the model outcomes.

---

[*]This research was funded in part by the NSF under grants DMR-1709351, IIS-1633387, IIS-1633724, and IIS-1815238, by the DARPA SD2 program, and the Arnold and Mabel Beckman Foundation. The Titan Xp used for this research was donated by the NVIDIA Corporation.

Feature influence methods can focus on the influence of a feature taken over all instances in the training or test set [7, 2], or on the local feature influence on a single *individual* item of the training or test set [25, 20] (both of which are different than the influence of a specific training instance on a model's parameters [16]). Both the global perspective given by considering the impact of a feature on all training and/or test instances as well as the local, individual perspective can be useful to investigate the fairness of a machine learning model. Consider, for example, the question of fairness in an automated hiring decision: determining the indirect influence of gender on all test outcomes could help us understand whether the system had disparate impacts overall, while an individual-level feature audit could help determine if a specific person's predictions were due in part to their gender.[2]

**Our Work.** In this paper we present a general technique to perform both global and individual-level indirect influence audits. Our technique is *modular* – it solves the indirect influence problem by reduction to a *direct influence* problem, allowing us to benefit from existing techniques.

Our key insight is that *disentangled representations* can be used to do indirect influence computation. The idea of a disentangled representation is to learn independent factors of variation that reflect the natural symmetries of a data set. This approach has been very successful in generating representations in deep learning that can be manipulated while creating realistic inputs [3, 4, 9, 17, 26]. Disentanglement has recently been shown to be a useful property for learning and evaluating fair machine learning models [6, 18]. Related methods use competitive learning to ensure a representation is free of protected information while preserving other information [8, 21].

In our context, the idea is to *disentangle* the influence of the feature whose (indirect) influence we want to compute. By doing this, we obtain a representation in which we can manipulate the feature directly to estimate its influence. Our approach has a number of advantages. We can connect indirect influence in the native representation to direct influence in the disentangled representation. Our method creates a *disentangled model*: a wrapper to the original model with the disentangled features as inputs. This implies that it works for (almost) any model for which direct influence methods work, and also allows us to use any future developed direct influence model.

Specifically, our disentangled influence audits approach provides the following contributions:

1. Theoretical and experimental justification that the disentangled model and associated disentangled influence audits we create provides an accurate indirect influence audit of complex, and potentially black box, models.

2. Quality measures, based on the error of the disentanglement and the error of the reconstruction of the original input, that can be associated with the audit results.

3. An indirect influence method that can work in association with both global and individual-level feature influence mechanisms. Our disentangled influence audits can additionally audit continuous features and image data; types of audits that were not possible with previous indirect audit methods (without additional preprocessing).

## 2  Our Methodology

### 2.1  Theoretical background

Let $P$ and $X$ denote sets of attributes with associated domains $\mathcal{P}$ and $\mathcal{X}$. $P$ represents *features of interest*: these could be protected attributes of the data or any other features whose influence we wish to determine. For convenience we will assume that $\mathcal{P}$ consists of the values taken by a single feature – our exposition and techniques work more generally. $X$ represents other attributes of the data that may or may not be influenced by features in $P$. An *instance* is thus a point $(p, x) \in \mathcal{P} \times \mathcal{X}$. Let $Y$ denote the space of *labels* for a learning task ($Y = \{+1, -1\}$ for binary classification or $\mathbb{R}$ for regression).

**Disentangled representation.** Our goal is to find an alternate representation of an instance $(p, x)$. Specifically, we would like to construct $x' \in \mathcal{X}'$ that represents all factors of variation that are *independent* of $P$, as well as an invertible mapping $f$ such that $f(p, x) = (p, x')$. We will refer to the original domain as $\mathcal{D} = \mathcal{P} \times \mathcal{X}$ and the associated new domain as $\mathcal{D}' = \mathcal{P} \times \mathcal{X}'$.

**Direct and indirect influence.** Given a model $M : \mathcal{D} \to Y$, a *direct influence* measure quantifies the degree to which any particular feature influences the outcome of $M$ on a specific input. In our experiments, we use the SHAP values proposed by [20] that are inspired by the Shapley values in game theory, but our framework applies to an arbitrary direct influence function $\phi$. For a model $M$ and input $x$, the influence of feature $p$ is defined by SHAP as [20, Eq. 8] $\phi_p(M, x) = \sum_{z \subseteq x} \frac{|z|!(n-|z|-1)!}{n!} [M_x(z) - M_x(z \setminus p)]$ where $\|z\|$ denotes the number of nonzero entries in $z$, $z \subseteq x$ is a vector whose nonzero entries are a subset of the nonzero entries in $x$, $z \setminus p$ denotes $z$ with the feature $p$ set to zero, and $n$ is the number of features. Finally, $M_x(z) = E[M(z)|z_S]$, the conditional expected value of the model subject to fixing all the nonzero entries of $z$ ($S$ is the set of nonzero entries in $z$).

*Indirect influence* attempts to capture how a feature might influence the outcome of a model even if it is not explicitly represented in the data, i.e its influence is via *proxy* features. The above direct influence measure cannot capture these effects because the information encoded in protected feature $p$ might be retained in other features even if $p$ is removed. We propose the following definition of indirect influence via a reduction to direct influence:

**Definition 1** (Indirect Influence). The indirect influence of a feature $p$ on the prediction of a model $M(p, x)$ is the direct influence $\phi_p$ of $p$ on the prediction of the disentangled model $(M \circ f^{-1})(p, x')$:

$$\mathcal{II}_p(M, (p, x)) = \phi_p(M \circ f^{-1}, (p, x')) \tag{1}$$

Equation (1) states that the indirect influence of $p$ is the direct influence of $p$ when considering $M$ as acting on the disentangled representation instead of the original features. Whereas direct influence measures the sensitivity of $M$ to changes in $p$ independent of all other features, indirect influence also considers how $p$ influences the prediction of $M$ through proxy features for $p$. Hence, indirect influence is inherently specific to a data distribution and should be interpreted with respect to the joint distribution for $(p, x)$ observed during training.[3] Here, a proxy for $p$ consists of a set of features $S$ and a function $g$ that *predicts* $p$: i.e such that $g(x_S) \simeq p$. Note that if there are no features that can predict $p$, then the indirect and direct influence of $p$ are the same (because the only proxy for $p$ is itself).

**Dealing with errors.** In practice, it might not be possible to perfectly learn the invertible mapping $f$ from $(p, x)$ to $(p, x')$. In particular, assume that our decoder function is some $g \neq f^{-1}$. While we do not provide an explicit formula for the dependence of the influence function parameters, we note that it is a linear function of the predictions, and so we can begin to understand the errors in the influence estimates by looking at the behavior of the predictor with respect to $p$.

Model output can be written as $\hat{y} = (M \circ g)(p, x')$. Recalling that $g(p, x') = (p, \hat{x})$, the partial derivative of $\hat{y}$ with respect to $p$ can be written as $\frac{\partial \hat{y}}{\partial p} = \frac{\partial (M \circ g)}{\partial p} = \frac{\partial M}{\partial \hat{x}} \frac{\partial \hat{x}}{\partial p} + \frac{\partial M}{\partial p} = \frac{\partial M}{\partial \hat{x}} \frac{\partial \hat{x}}{\partial x'} \frac{\partial x'}{\partial p} + \frac{\partial M}{\partial p}$. Consider the term $\frac{\partial x'}{\partial p}$. If the disentangled representation is perfect, then this term is zero (because $x'$ is unaffected by $p$), and therefore we get $\frac{\partial \hat{y}}{\partial p} = \frac{\partial M}{\partial p}$ which is as we would expect. If the reconstruction is perfect (but not necessarily the disentangling), then the term $\frac{\partial \hat{x}}{\partial x'}$ is 1. What remains is the partial derivative of $M$ with respect to the latent encoding $(x', p)$.

## 2.2 Implementation

Our overall process requires two separate pieces: 1) a method to create disentangled representations, and 2) a method to audit direct features. In most experiments in this paper, we use methods related to adversarial autoencoders [22] to generate disentangled representations, and Shapley values from the `shap` technique for auditing direct features [20] (as described above in Section 2.1).

We train a disentangled representation to estimate $(p, x')$ for each feature of interest $p$. This allows us to compute representations with only two factors in a supervised manner, avoiding many of the practical issues affecting methods for learning disentangled representations [19]. A key limitation of this approach is that while easier to train, it potentially requires one to train many disentangled representations. This means the technique may be most useful in domains such as fairness where we

care specifically about the impact of one or a small collection of distinguished features that may or may not be directly used as inputs to the model.

**Disentangled representations via adversarial autoencoders** We create disentangled representations by training three separate neural networks, which we denote $f$, $g$, and $h$ (see Figure 1). Networks $f$ and $g$ are autoencoders: the image of $f$ has lower dimensionality than the domain of $f$, and the training process seeks for $g \circ f$ to be an approximate identity, through gradient descent on the reconstruction error $||(g \circ f)(x) - x||$.

Unlike regular autoencoders, $g$ is also given direct access to the protected attribute. *Adversarial* autoencoders [22], in addition, use an ancillary network $h$ that attempts to recover the protected attribute from the image of $f$, *without access to $p$ itself*. (Note the slight abuse of notation here: $h$ is assumed not to have access to $p$, while $g$ does have access to it.) During the training of $f$ and $g$, we seek to reduce $||(g \circ f)(x) - x||$, but also to *increase the error* of the discriminator $h \circ f$. The optimization process of $h$ tries to recover the protected attribute from the code generated by $f$. ($h$ and $f$ are the *adversaries*.) When the process converges to an equilibrium, the code generated by $f$ will contain no information about $p$ that is useful to $h$, but $g \circ f$ still reconstructs the original data correctly: $f$ disentangles $p$ from the other features.

The loss functions used to codify this process are $\mathcal{L}_{\text{Enc}} = \text{MSE}(x, \hat{x}) - \beta\,\text{MSE}(p, \hat{p})$, $\mathcal{L}_{\text{Dec}} = \text{MSE}(x, \hat{x})$, and $\mathcal{L}_{\text{Disc}} = \text{MSE}(p, \hat{p})$, where MSE is the mean squared error and $\beta$ is a hyperparameter determining the importance of disentanglement relative to reconstruction. When $p$ is a binary feature, $\mathcal{L}_{\text{Enc}}$ and $\mathcal{L}_{\text{Disc}}$ are adjusted to use binary cross entropy loss between $p$ and $\hat{p}$.

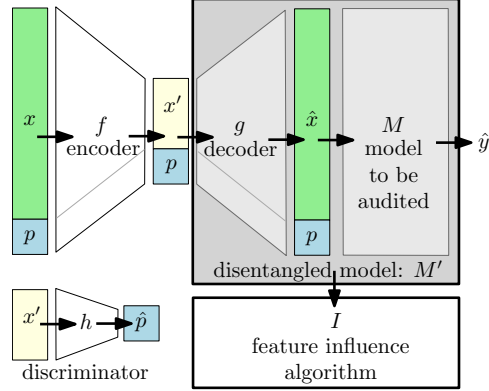

Figure 1: System diagram when auditing the indirect influence of feature $p$ on the outcomes of model $M$ for instance $(p, x)$. The autoencoder $g \circ f$ learns the instance $(p, x)$ as a function of independent factors $(p, x')$. The independence of $p$ and $x'$ is enforced by the adversary $h$. The disentangled representation $(p, x')$ is the input for the disentangled model $M' = M \circ g$ which is audited using the direct influence algorithm $I$.

**Disentangled feature audits** Concretely, our method works as follows, where the variable names match the diagram in Figure 1:

DISENTANGLED-INFLUENCE-AUDIT$(X, M)$

1   **for** $p$ **in** FEATURES(X)
2        $(f, g, h) = $ DISENTANGLED-REPRESENTATION$(X, p)$ // ($h$ is not used)
3        $M' = g \circ M$
4        $X' = \{f(x)\ \textbf{for}\ x\ \textbf{in}\ X\}$
5        $\text{SHAP}_p = $ DIRECT-INFLUENCE$(X', p, M')$
6   **return** $\{\text{SHAP}_p\ \textbf{for}\ p\ \textbf{in}\ \text{FEATURES}(X)\}$

While we use shap in our implementations, our framework applies to other direct influence functions as well. We note here one important difference in the interpretation of disentangled influence values when contrasted with regular Shapley values. Because the influence of each feature is determined on a *different* disentangled model, the scores we get are not directly interpretable as a partition of the model's prediction. For example, consider a dataset in which feature $p_1$ is responsible for 50% of the direct influence, while feature $p_2$ is a perfect proxy for $p_1$, but shows 0% influence under a direct audit. Relative judgments of feature importance remain sensible.

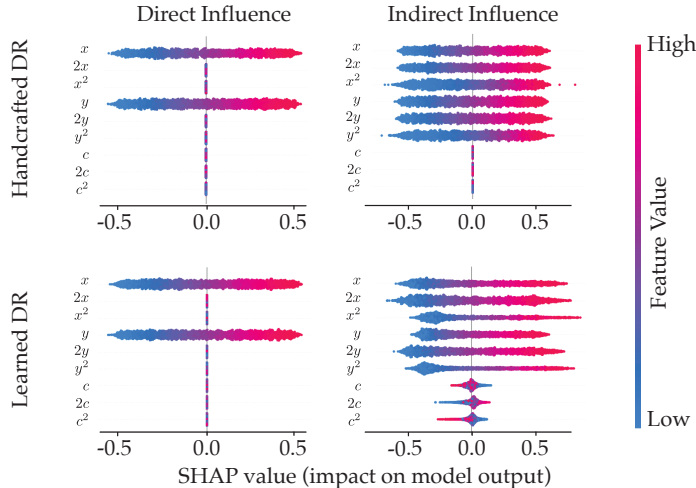

Figure 2: Synthetic $x + y$ data direct `shap` (left) and indirect (right) feature influences using a handcrafted (top row) or learned disentangled representation (bottom row). In each plot, a point represents the influence of a feature on a single prediction. Points far from the center indicate high importance, so a row with many points far from the center indicates a feature which is deemed important. As expected, the direct influence method `shap` reports that the only important features are $x$ and $y$, but our methods capture that $x$ and $y$ are perfect proxies for $2x, 2y, x^2, y^2$ so these features have equal indirect influence.

## 3 Experiments

In this section, we will assess the extent to which disentangled influence audits are able to identify sources of indirect influence to a model and quantify its error. All data and code[4] for the described method and below experiments is available in the Supplementary Materials.

### 3.1 Synthetic $x + y$ Regression Data

In order to evaluate whether the indirect influence calculated by the disentangled influence audits correctly captures all influence of individual-level features on an outcome, we will consider influence on a simple synthetic $x + y$ dataset. It includes 5,000 instances of two variables $x$ and $y$ drawn independently from a uniform distribution over $[0, 1]$ that are added to determine the label $x + y$. It also includes proxy variables $2x, x^2, 2y$, and $y^2$. A random noise variable $c$ is also included that is drawn independently of $x$ and $y$ uniformly from $[0, 1]$. The model we are auditing is a handcrafted model that contains no hidden layers and has fixed weights of 1 corresponding to $x$ and $y$ and weights of 0 for all other features (i.e., it directly computes $x + y$). We use `shap` as the direct influence delegate method [20].[5]

In order to examine the impact of the quality of the disentangled representation on the results, we considered both a handcrafted disentangled representation and a learned one. For the former, nine unique models were handcrafted to disentangle each of the nine features perfectly (see Supplementary Materials for details). The learned disentangled representation is created according to the adversarial autoencoder methodology described in more detail in the previous section.

The results for the handcrafted disentangled representation (top of Figure 2) are as expected: features $x$ and $y$ are the only ones with direct influence, all $x$ or $y$ based features have the same amount of indirect influence, while all features including $c$ have zero influence. Using the learned disentangled representation introduces the potential for error: the resulting influences (bottom of Figure 2) show more variation between features, but the same general trends as in the handcrafted test case.

Additionally, note that since `shap` gives influence results per individual instance, we can also see that (for both models) instances with larger (or, respectively, smaller) $2x$ or $2y$ values give larger (respectively, smaller) results for the label $x + y$, i.e., have larger absolute influences on the outcomes.

### 3.1.1 Error Analyses

There are two main sources of error for disentangled influence audits: error in the reconstruction of the original input $x$ and error in the disentanglement of $p$ from $x'$ such that the discriminator is able to accurately predict some $\hat{p}$ close to $p$. We will measure the former error in two ways. First, we will consider the *reconstruction error*, which we define as $x - \hat{x}$. Second, we consider the *prediction error*, which is $M(x) - M(\hat{x})$ - a measure of the impact of the reconstruction error on the model to be audited. Reconstruction and prediction errors close to 0 indicate that the disentangled model $M'$ is similar to the model $M$ being audited. Common measures for disentanglement such as the mutual information gap (MIG) do not apply well to our method since we disentangle the features one at a time, as opposed to simultaneously [5]. We measure the *disentanglement error*, as $\frac{1}{n}\sum_{i=1}^{n}(p - \hat{p})^2/var(p)$ where $var(p)$ is the variance of $p$. A disentanglement error of below 1 indicates that information about that feature may have been revealed, i.e., that there may be indirect influence that is not accounted for in the resulting influence score. In addition to the usefulness of these error measures during training time, they also provide information that helps us to assess the quality of the indirect influence audit, including at the level of the error for an individual instance.

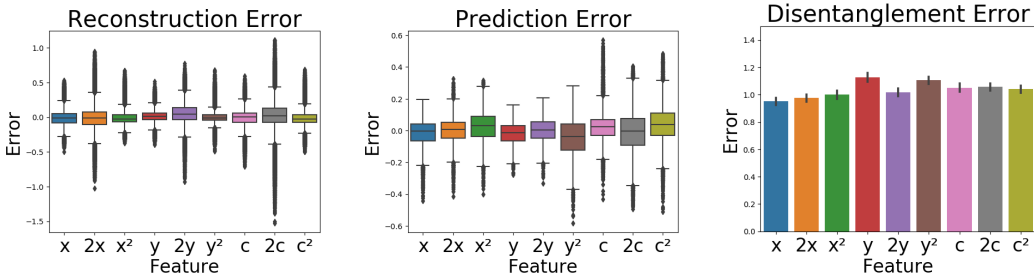

Figure 3: Errors on the synthetic $x + y$ data for the reconstruction error (left) when taken across influence audits for each feature, prediction error (middle), and disentanglement error (right). Optimal is reconstruction error and prediction error of 0 for all features (indicating no errors in autoencoding), and disentanglement error of 1 for all features (indicating $p$ and $x'$ are independent).

These influence experiments on the $x + y$ dataset demonstrate the importance of a good disentangled representation to the quality of the resulting indirect influence measures, since the handcrafted zero-error disentangled representation clearly results in more accurate influence results. Each of the error types described above are given for the learned disentangled representation in Figure 3. While most features have reconstruction and prediction errors close to 0 and disentanglement errors close to 1, a few features also have some far outlying instances. For example, we can see that the $c, 2c,$ and $c^2$ variables have high prediction error on some instances, and this is reflected in the incorrect indirect influence that they're found to have on the learned representation for some instances.

### 3.2 dSprites Image Classification

The second synthetic dataset is the **dSprites** dataset commonly used in the disentangled representations literature to disentangle independent factors that are sources of variation [23]. The dataset consists of $737,280$ images ($64 \times 64$ pixels) of a white shape (a square, ellipse, or heart) on a black background. The independent latent factors are $x$ position, $y$ position, orientation, scale, and shape. The images were downsampled to $16 \times 16$ resolution and the half of the data in which the shapes are largest were

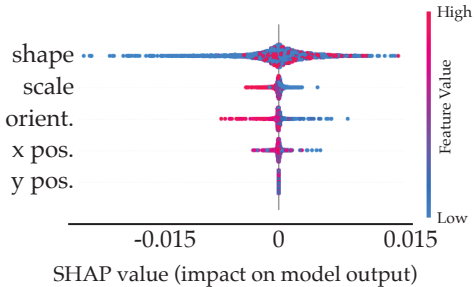

Figure 4: dSprites data indirect latent factor influences on a model predicting shape.

used. The binary classification task is to predict whether the shape is a heart. A good disentangled representation should be able to separate the shape from the other latent factors.

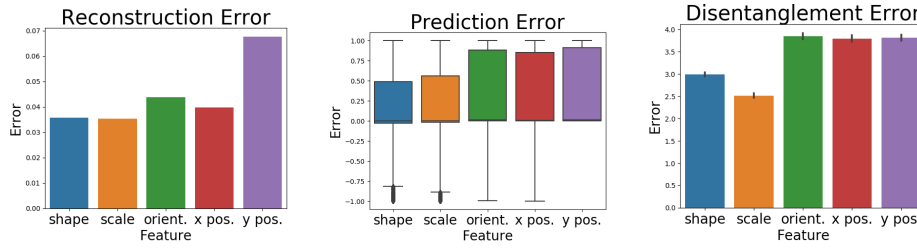

Figure 5: The mean squared reconstruction error (left), absolute prediction error (middle), and absolute disentanglement error (right) of the latent factors in the dSprites data under an indirect influence audit. Optimal is reconstruction error and prediction error of 0, and disentanglement error of 1. We see that the quality of the disentangled representation varies for the dSprites data.

In this experiment we seek to quantify the indirect influence of each latent factor on a model trained to predict the shape from an image. Since `shape` is the label and the latent factors are independent, we expect the feature `shape` to have more indirect influence on the model than any other latent factor. Note that a direct influence audit is impossible since the latent factors are not themselves features of the data. Model and disentangled representation training information can be found in the Supplementary Material.

The indirect influence audit, shown in Figure 4, correctly identifies `shape` as the most important latent factor, and also correctly shows the other four factors as having essentially zero indirect influence. However, the audit struggles to capture the extent of the indirect influence of `shape` since the resulting `shap` values are small.

The associated error measures for the dSprites influence audit are shown in Figure 5. We report the reconstruction error as the mean squared error between $x$ and $\hat{x}$ for each latent factor. The prediction error is the difference between $M(x)$ and $M(\hat{x})$ of the model's estimate of the probability the shape is a heart. While the reconstruction errors are relatively low (less than 0.05 for all but $y$ position) the prediction error and disentanglement errors are high. A high prediction error indicates that the model is sensitive to the errors in reconstruction and the indirect influence results may be unstable, which may explain the low `shap` values for shape in the indirect influence audit.

## 3.3   Adult Income Data

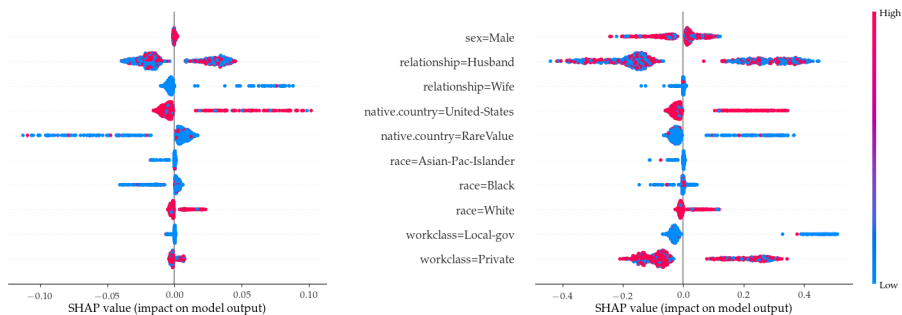

Figure 6: Ten selected features for Adult dataset. Direct (left) and indirect (right) influence are shown. For all features, see Supplemental Material. Low values indicate a one-hot encoded feature is `false`. Features with many points far from the center (shown here using width of a cluster) are identified as being of high importance. These results indicate that features `sex=Male`, `relationship=Husband` and `workclass=Private` may be used by the model via proxy variables since they have higher indirect influence than direct influence.

Finally, we will consider a real-world dataset containing **Adult Income** data that is commonly used as a test case in the fairness-aware machine learning community. The Adult dataset includes 14 features describing type of work, demographic information, and capital gains information for individuals from the 1994 U.S. census [27]. The classification task is predicting whether an individual makes more or less than $50,000 per year. Preprocessing, model, and disentangled representation training information are included in the Supplementary Material.

Direct and indirect influence audits on the Adult dataset are given in Figure 6 and in the Supplementary Material. While many of the resulting influence scores are the same in both the direct and indirect cases, the disentangled influence audits finds substantially more influence based on `sex` than the direct influence audit - this is not surprising given the large influence that sex is known to have on U.S. income. Other important features in a fairness context, such as nationality, are also shown to have indirect influences that are not apparent on a direct influence audit. The error results (Figure 7 and the Supplementary Material) indicate that while the error is low across all three types of errors for many features, the disentanglement errors are higher (further from 1) for some rare-valued features. This means that despite the indirect influence that the audit did find, there may be additional indirect influence it did not pick up for those features.

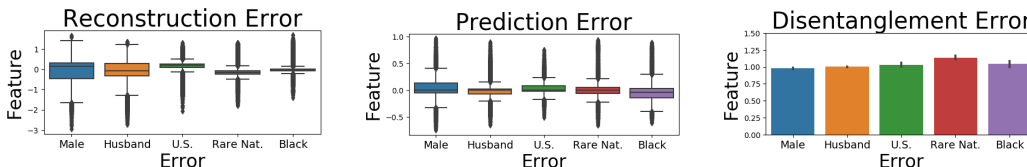

Figure 7: The reconstruction error (left), prediction error (middle), and disentanglement error (right) of selected Adult Income features under an indirect influence audit. Optimal is a reconstruction error and prediction error of 0, and a disentanglement error of 1 for all features. See the Supplementary Material for the complete figure.

## 3.4 Comparison to Other Methods

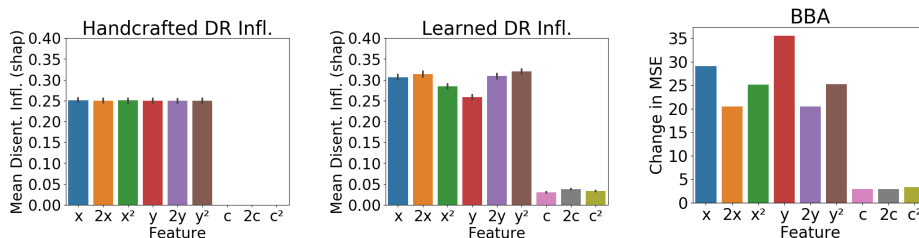

Figure 8: Comparison on the synthetic $x + y$ data of the disentangled influence audits using the handcrafted (left) or learned (middle) disentangled representation with the BBA approach of [2] (right). According to our definition of indirect influence and using `shap` the features $x, y, 2x, 2y, x^2, y^2$ should have the same influence and $c, 2c, c^2$ should have no influence.

Here, we compare the disentangled influence audits results to results on the same datasets and models by the indirect influence technique introduced in [2], which we will refer to as BBA (black-box auditing).[6] However, this is not a direct comparison, since BBA is not able to determine feature influence for individual instances, only influence for a feature taken over all instances. In order to compare to our results, we will thus take the mean over all instances of the absolute value of the per feature disentangled influence. BBA was designed to audit classifiers, so in order to compare to the results of disentangled influence audits we will consider the obscured data they generate as input into our regression models and then report the average change in mean squared error for the case of the synthetic $x + y$ data. (BBA cannot handle dSprites image data as input.)

A comparison of the disentangled influence and BBA results on the synthetic $x + y$ data shown in figure 8 shows that all three variants of indirect influence are able to determine that the $c, 2c, c^2$ variables have comparatively low influence on the model. The disentangled influence with a handcrafted disentangled representation shows the correct indirect influence of each feature, while the learned disentangled representation influence is somewhat more noisy, and the BBA results suffer from relying on the mean squared error (i.e., the amount of influence changes based on the feature's value).

Figure 9 shows the mean absolute disentangled influence per feature on the x-axis and the BBA influence results on the y-axis. The features with large disentangled influence and low BBA score

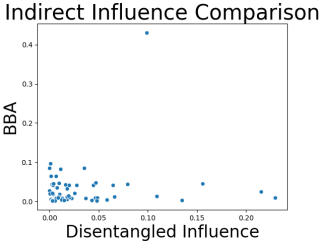

Figure 9: Comparison on the Adult data of the disentangled influence audits versus the BBA indirect influence approach of [2]. Our disentangled feature audits identifies more plausible, potentially-influential features than BBA. See text for details.

are `marital.status=Married-civ-spouse` and `relationship=Husband`. BBA can only detect influence present in pairwise dimensions, and not more complex high-dimensional correlations; perhaps this is why marital status is found to have such a large influence by the disentangled influence audit and not by BBA. The feature with large BBA score is `age`, and the reconstruction error on the disentangled influence audit for that feature indicates that the audit may not have picked up the full influence of that feature. It's clear that overall the disentangled influence audits technique is much better able to find features with possible indirect influence on this dataset and model: most of the BBA influences are clustered near zero, while the disentangled influence values provide more variation and potential for insight.

**SHAP vs. LIME** In Section 3.3, we use SHAP audits as our direct and indirect audit sources. As Lee and Lundberg argue, the SHAP values are, in essence, a variation of the LIME method [20], one that provides weights for samples and features that are consistent with the Shapley value formulation from game theory. As a result, the audits for LIME are not fundamentally different than those for SHAP; we provide them in the Supplementary Material.

## 4 Discussion and Conclusion

In this paper, we introduce the idea of disentangling influence: using the ideas from disentangled representations to allow for indirect influence audits. We show via theory and experiments that this method works across a variety of problems and data types including classification and regression as well as numerical, categorical, and image data. The methodology allows us to turn any future developed direct influence measures into indirect influence measures. In addition to the strengths of the technique demonstrated here, disentangled influence audits have the added potential to allow for multidimensional indirect influence audits that would, e.g., allow a fairness audit on both race and gender to be performed (without using a single combined race and gender feature [10]). We hope this opens the door for more nuanced fairness audits.

## Footnotes

[2]While unrelated to feature influence, the idea of *recourse* [28] also emphasizes the importance of individual-level explanations of an outcome or how to change it.

[3]See the related perspective of disentangled representations as recovering symmetries in underlying *world states*, which directly inspired our approach [13].

[4]Code is available at: `https://github.com/charliemarx/disentangling-influence`

[5]This method is available via `pip install shap`. See also: `https://github.com/slundberg/shap`

[6]This method is available via `pip install BlackBoxAuditing`. See also: `https://github.com/algofairness/BlackBoxAuditing`

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
