[Supplementary Material · supplementary.pdf]

# Disentangling Influence: Using disentangled representations to audit model predictions

## Supplementary Materials

## 1 Implementation Details

**Synthetic $x + y$ model and disentangled representation information**     In both our synthetic experiments with handcrafted and trained disentangled representations we audit a model with no hidden layers that computes $x + y$ exactly from the features $x$ and $y$.

The handcrafted disentangled representation is created to map the features with no error. Suppose for example the protected feature, denoted $p$, was one of the features based on $y$ (one of $y, 2y, y^2$). The disentangled representation used in this case would be $([x, c], [p])$. Here, we see that p will fully reveal the information relating to all of the features based on y, and $X' = [x, c]$ does not reveal any information about the protected feature. Thus, this representation satisfies the independence and preservation of information requirements. The decoder then maps this vector back to the original feature vector $(x, 2x, x^2, y, 2y, y^2, z, 2z, z^2)$, in the natural way. If for example $p = y^2$, the decoder first computes $\sqrt{p}$ to calculate $y$, then uses this to compute $2y$. All features relating to $x$ and $z$ are computed from $x$ and $z$ in the natural way as well.

In the disentangled representation we train the encoder, decoder and discriminator each have two hidden layers of 10 hidden units each. We use a 4 dimensional latent vector. All layers in each model have ReLU activations except for the last layer of the decoder and discriminator which have sigmoid activations. We use $\beta = 0.5$ as the importance of disentanglement for the encoder. The minibatch size is 16 and we optimize for 10,000 train steps using SGD with a constant learning rate of 0.01.

**dSprites model and disentangled representation information**     The model we use to predict the shape from the image is a neural network with three layers of 128, 64, and 32 hidden units respectively, and achieves a $97\%$ prediction accuracy on a held out test set. The test set was randomly drawn as $20\%$ of the data. To generate the disentangled representation we use an encoder, decoder and discriminator each with a single hidden layer of 256, 256 and 64 hidden units respectively. We use a 16 dimensional latent vector. The minibatch size is 100 and we optimize for 10,000 train steps using SGD with a constant learning rate of 0.05. All layers in each model have ReLU activations except for the last layer of the decoder and discriminator which have sigmoid activations. We use $\beta = 1$ as the importance of disentanglement for the encoder.

**Adult Income preprocessing, model, and disentangled representation information**     During preprocessing, categorical features are one-hot encoded and numerical features are normalized to mean 0 and standard deviation 1. The "education_num" feature is dropped during preprocessing. For each categorical feature, values which occur in less than 1,000 instances are binned into "rare_value". We train a classifier for the "income>=50K" label with binary cross entropy loss and no hidden layers. The classifier achieves test loss of 0.326 and test accuracy of $84.9\%$.

To generate the disentangled representation we use an encoder, decoder and discriminator which each have two hidden layers with 25 and 12 hidden units respectively. We use a 10 dimensional latent vector. We use $\beta = 0.5$ as the importance of disentanglement for the encoder. The models are trained for 4000 train steps with minibatch sizes of 16, using SGD with a constant learning rate of 0.01. We used the canonical train/test split.

**Additional Information** All models for the synthetic $x + y$ and dSprites experiments were trained on a MacBook Pro (Early 2015) with a 2.7GHz Processor and 8 GB of RAM. The models for the adult experiments were trained on an NVIDIA Titan Xp GPU. Hyperparameters were chosen via experimentation. Only architectures containing 2 or fewer hidden layers were considered for models used to disentangle the data. The minibatch sizes tested were between 16 and 100, and learning rates between 0.01 and 0.1 were tested. In each experiment, we used at least 5 and no more than 15 evaluation runs.

# 2 Full results for Adult data set

## 2.1 Direct and Indirect Influence Results

Figure 1: The full influence results for the adult data direct (left) and indirect (right) feature influences.

 **2.2   Error Results**

## Disentanglement Error

The chart shows Error values (x-axis, 0.0 to 1.4) for various Features (y-axis):

- age
- workclass=?
- workclass=Federal-gov
- workclass=Local-gov
- workclass=Private
- workclass=RareValue
- workclass=Self-emp-inc
- workclass=Self-emp-not-inc
- workclass=State-gov
- education=10th
- education=11th
- education=Assoc-acdm
- education=Assoc-voc
- education=Bachelors
- education=HS-grad
- education=Masters
- education=RareValue
- education=Some-college
- education.num
- marital.status=Divorced
- marital.status=Married-civ-spouse
- marital.status=Never-married
- marital.status=RareValue
- marital.status=Separated
- marital.status=Widowed
- occupation=?
- occupation=Adm-clerical
- occupation=Craft-repair
- occupation=Exec-managerial
- occupation=Farming-fishing
- occupation=Handlers-cleaners
- occupation=Machine-op-inspct
- occupation=Other-service
- occupation=Prof-specialty
- occupation=RareValue
- occupation=Sales
- occupation=Tech-support
- occupation=Transport-moving
- relationship=Husband
- relationship=Not-in-family
- relationship=Other-relative
- relationship=Own-child
- relationship=Unmarried
- relationship=Wife
- race=Asian-Pac-Islander
- race=Black
- race=RareValue
- race=White
- sex=Male
- capital.gain
- capital.loss
- hours.per.week
- native.country=RareValue
- native.country=United-States

Reconstruction Error

## Prediction Error