[Reviews · NeurIPS 2019]

Reviewer 1



Originality: while somewhat similar to [Adler et al 2018], which they compare against, this submission does seem to make novel contributions both in their theoretical framing of the problem and in their practical solution, which feel like solid improvements. Quality: the submission seems technically sound, and the authors seem pretty honest about both strengths and weaknesses. I appreciated the error analyses and the demonstration on toy data (Figure 2), which shows that the method isn't perfect but still captures something important. I do have a few suggestions and points of criticism which I will list below. Clarity: the paper is mostly well-written and well-organized, though I have a few suggestions for improvements here too. Significance: So, I have mixed feelings here. I do feel like this paper provides one of the first mostly-fleshed-out solutions to an extremely important problem, and that the method it provides should work on datasets which are simple enough. However, I worry that on the majority of real-world datasets (e.g. very high-dimensional tabular data or big images), the autoencoder training step is just not going to work -- discriminative models are a lot easier to train than generative ones. In that case, something like TCAV [https://arxiv.org/pdf/1711.11279.pdf] might be a lot more useful for auditing. So I think that limits the significance of this work. But it still could be an important building block, and it does provide quality measures that ensure it doesn't "fail silently." --- Updates based on author feedback and reviewer discussion: Overall, I was happy with the author feedback and think the rewrites/reorganizations which were planned will strengthen the paper. Accordingly, I am strengthening my score. I do want to echo one of R2's points, though, which is that -- although this is a "local" explanation method -- the disentanglement objective is in some sense "global." While I don't think this makes your method equivalent to demographic (dis)parity-based approaches (since one can have demographic parity but still have significant indirect influence by sensitive attributes that happens to average out to zero), it might be worth considering whether there is any notion of "local" disentanglement you could use instead. However, that's pretty difficult, and overall I feel this paper will help advance the field and enhance our ability to address important problems.

Reviewer 2



The authors are interested in computing the influence of a feature p on the prediction of model M, both in terms of individual predictions and global trends. While previous work in interpreting model predictions has focused on computing feature importance via saliency maps and sensitivity analysis of the model to input perturbations, the authors are concerned with the total influence of p on the model predictions, including through model sensitivity to features besides p which nonetheless serve as proxies for p. I believe this is a useful direction of research in model interpretability. I also believe that the idea to learning an invariant/factorized intermediate representation is interesting. However, in my assessment the submission is not up to the standard required for publication. There are improvements to be made in terms of properly defining and motivating the problem, reducing formalisms that are not used by the implemented method, and improving the experiments. I offer some concrete suggestions below. The experimental justification for the proposed method in particular requires some attention (again, I will elaborate below). Put briefly, the authors deviate from the standard approach to modeling DSprites with disentanglement priors (e.g., in data dimentionality, architectures, and metrics) and their baseline comparisons are not enlightening. As an overall impression, I felt that the paper was not very clearly written, and the claim from the abstract that "our method is more powerful than existing methods for ascertaining feature influence" was not substantiated. I hope that the authors are not too discouraged by this feedback, since I suspect there is a useful contribution to be had in this direction given further empirical work and a rewrite of the paper. If the authors are interested in more of a fairness spin to this work, I would suggest including at least one additional tabular dataset besides Adult.

Reviewer 3



Overall Comments One significant difficulty in a lot of feature importance/significance assessments is the issue of multicollinearity, which this paper notes as proxy features. The literature on disentangling could be potentially useful in this setting. It is interesting that the paper leverages this connection to help address this issue. In my opinion, this is an interesting connection and one that has not been fully explored in the literature. Originality As far as I am aware, this work is the first to combine work from the disentangling literature with the SHAP method for interpretability. One could contend that disentangling is essentially clustering and that feature selection (which is what SHAP does) on top of clustering is not novel; however, I would still say that this work is. Quality I consider this paper a comprehensive assessment of the thesis that the paper sets forth, i.e., disentangling helps to better interpret when there is proxy influence. Overall the paper is well written with the exception of section 2.1. I also commend the authors for overall clarity in general. I note some points about section 2.1 later in this review. Significance. The issue of dealing with proxy features and multicollinearity is a timely problem and particularly potent one in the fairness + ml literature. Here it is important to properly disambiguate the impact/influence of a protected attribute like race etc on the outcome. The goal of the disentangled influence work directly addresses this issue. My biggest issue with this work is with the disentangled representations portion. It is hard to judge the output of these methods and to properly assess whether representations are disentangled or to even know what it means for something to be disentangled. I'll return to this later in my review. In spite of this, I contend that this is still a useful contribution on the part of the authors. Some gripes - Disentangled representations: I have followed this literature lately, and it seems there is significant doubt whether most of the methods in this area works. In particular, the recent ICML paper: https://arxiv.org/abs/1811.12359.pdf. First, it is not clear what kinds of dataset it is possible to learn a disentangled representations for. Second, even if possible, the field is not clear on what metrics are useful for assessing this disentanglement. If such disentanglement were possible and verifiable, I agree with the authors that marrying feature selection with disentangled representations is a significant undertaking that should help solve multicollinearity issues. This is my biggest gripe with this line of work; however, since several of the previous papers were accepted at top conferences, I think this work is interesting as a proof of concept of what is possible along these lines. After the somewhat philosophical rant, this section pertains to some methodological choices the paper makes: - Why SHAP? It seems to me that if one has disentangled representations, and if the representations are 'truly' disentangled, then a pipeline with any reasonable feature selection method should work here? Here is what I mean, right now the authors propose disentangled representations + SHAP, I argue that disentangled representations + {SHAP, LIME, Correlation Coef, etc} should suffice. I do understand that SHAP has been shown to provably subsume LIME and a whole host of other methods. - Why not compare your encoder + decoder scheme with betaVAE or the some other class of methods that can learn disentangled representations? - Section 2.1: I read this section a few times, and I understand that it is supposed to provide theoretical justification for your work, however, I don't quite see how it fits with the rest of the paper. Proposition 1 follows pretty straightforwardly from the definition of a disentangled representation, so am not sure why it is needed? - The synthetic example is (section 3.1) is very good, and indeed demonstrates that disentangled representations + {SHAP} can capture proxy influence from variables 2x y^2 etc. - Does the paper address disentanglement of p, the sensitive attribute, in general or for all variables. I was confused on this issue. Does the method seek a representation that where p has been 'removed' or does the method seek to factor all 'features' in the input into 'independent' components? If it is just to remove the variable p, then it makes sense to not test against disentangled representation methods more generally. - The paper's section on error analyses gets to the core of my biggest issue with the disentanglement literature. Do the authors envisions that people will perform this kind of error analyses before using their method? Overall, I consider this work to be interesting since it bridges two interesting areas of research. The authors also perform pretty comprehensive assessment of the proposed method on real datasets. Update I have read the author rebuttal, and it indeed clarified a some questions that were brought up as part of the review. I am planing to maintain my accept rating for this work.

[Author Response · NeurIPS 2019]

We thank the reviewers for their incredibly detailed and thoughtful reviews. We really appreciate the time you put into reading and thinking about our work. The top issues we understood from the reviews were 1) lack of clarity in Section 2.1 on the usefulness of the [Higgins et al.] framework in this context and the formalization of the indirect influence definition, 2) lack of clarity about the implementation, and 3) concerns about the experimental section. Below, we include partial rewrites of Sections 2.1 and 2.2 that we will include directly in our revision and we hope clarify the first two of these issues (we describe planned improvements to the experimental section below). Note the indirect influence definition that is now more explicit about the use of the disentangled representations theory and the hopefully clearer explanation that the method uses a reductions framework.

Our primary goal is to leverage recent developments in disentangled representations to help solve the indirect influence problem for individuals, and not to supersede current work in disentangled representations. Since our goal is to compute individual influence scores, we cannot use global metrics such as demographic parity difference as reviewer 2 suggests. Given a sufficiently strong adversary, our disentanglement error metric reports high error when there is mutual information between $p$ and $x'$, and has low error when there is low mutual information between $p$ and $x'$, making it very related to the mutual information gap. We appreciate the reviewer's suggestion to report the mutual information gap as well, and will add this to our analyses.

In accordance with the suggestions of the reviewers, we intend to improve the experimental section in the following ways: 1) We will update the dSprites experiment to operate on the full $64 \times 64$ pixel images, using the standard CNN architectures in the literature. 2) We will add existing error metrics for our disentangled representations, including mutual information. 3) We will add the additional baseline of LIME, as well as more detailed exposition about how our method compares to the baselines.

We note that in the dSprites experiment our goal is to compute the influence of the latent factors on the predictions of a model trained only on the pixels, by learning how the pixels act as proxies for the latent factors. We are not trying to model relationships between the latent features themselves. It is convenient for verifying our method that these latent factors are independent, so that the ground-truth is easily interpretable. This independence does not compromise our experimental goals. Still, we agree that adding relationships between the latent factors would be an interesting extension.

We thank the reviewers for pointing out that we should emphasize the importance of indirect influence to fair machine learning and we will revise the introduction to further emphasize this application. We also appreciate the reviewers' links to other relevant papers and will incorporate these into our related work description as well as the detailed stylistic suggestions.

**Section 2.1 Partial Revision:**

**Definitions** We use the term *world state* [Higgins et al.] to represent the actual nature of the objects or people represented in the data, void of all of the errors and omissions incurred during observation. We say the world state consists of two independent factors of variation, $(p, x')$ which correspond to the protected and unprotected aspects of the world state respectively. The unprotected features $\mathbf{x}$ are generated from the world state by an observation process $b : W \to \mathcal{X}$ so that $b(p, \mathbf{x'}) = (p, \mathbf{x})$. Furthermore, let $b_i$ be the function such that $b_i(p, \mathbf{x'})$ is the observation of only $x_i \in \mathbf{x}$. To provide generality to multiple forms of direct influence, we assume an arbitrary direct influence function $\mathcal{DI} : \mathcal{X} \times \hat{\mathcal{Y}} \to \mathbb{R}$. We formulate our implementations using SHAP as our notion of direct influence, but our framework is general and is compatible with other common local interpretability methods such as LIME and gradient based methods. We propose the following definition of indirect influence via a reduction to direct influence:

$$\mathcal{II}_p[M(p, x)] \coloneqq \mathcal{DI}_p[(M \circ b)(p, x')]$$

The above states that the indirect influence of $p$ is the direct influence of $p$ when considering the model as acting on world states instead of features. Whereas direct influence measures the sensitivity of a model to changes in each feature independently, indirect influence attempts to model how proxies for $p$ change along with $p$. Note that indirect influence is inherently specific to a data distribution, since our goal is to understand proxy relationships between features. All indirect influence audits should then be interpreted as with respect to the dependence structures observed during training.

**Section 2.2 Partial Revision:**

**Implementation** We train a disentangled representation to estimate $(p, x')$ for each feature of interest $p$. This allows us to compute representations with only two factors in a supervised manner, avoiding many of the issues in the current disentangled representations literature noted by [Locatello et al.]. A key limitation of this approach is that while easier to train, it potentially requires one to train many disentangled representations. This means the technique may be most useful in domains such as fairness where we care specifically about the impact of one or a small collection of distinguished features that may or may not be directly used as inputs to the model.

[Meta-Review · NeurIPS 2019]

We thank the authors for an interesting rebuttal and paper which sparked discussion among the reviewers. The reviewers have updated their responses accordingly. Although there was discrepancy in the scores, it was agreed that the paper is well-motivated and presents an important problem domain. The main issue with this work was whether one can really learn this disentangled representation in practice. However, the paper presents an excellent proof of concept.